# Study of Food Intake and Physical Activity Patterns in the Working Population of the Uruguayan State Electrical Company (UTE): Design, Protocol and Methodology

**DOI:** 10.3390/nu13103545

**Published:** 2021-10-09

**Authors:** Maria Mercedes Medina-Vadora, Cecilia Severi, Carlos Lecot, Maria Dolores Ruiz-Lopez, Angel Gil

**Affiliations:** 1Department of Nutrition and Food Sciences, Faculty of Pharmacy, University of Granada, 18071 Granada, Spain; lic.mercedesmedina@gmail.com; 2Department of Preventive Medicine, School of Medicine, Universidad de la República Oriental del Uruguay (UdelaR), Montevideo 11800, Uruguay; severi.cecilia@gmail.com; 3Uruguayan Society of Collective Health (SUSAC), Montevideo 11800, Uruguay; 4Department of Occupational Health, National Administration of Power Plants and Electric Transmissions (UTE), Montevideo 11800, Uruguay; clecot@ute.com.uy; 5Biomedical Research Center, Institute of Nutrition and Food Technology “José Mataix”, University of Granada, 18100 Granada, Spain; agil@ugr.es; 6Iberoamerican Nutrition Foundation (FINUT), Armilla, 18016 Granada, Spain; 7Department of Biochemistry and Molecular Biology II, University of Granada, 18071 Granada, Spain; 8CIBEROBN (Physiopathology of Obesity and Nutrition CB12/03/30038), Instituto de Salud Carlos III (ISCIII), 28029 Madrid, Spain

**Keywords:** food intake, lifestyle habits, physical activity, food consumption, workers nutritional state, food intake patterns, nutritional surveys

## Abstract

Noncommunicable diseases are the main cause of death globally, and most are potentially preventable; they are long term diseases and generally evolve slowly. In Uruguay 64.9% of the population between 25 and 64 years of age are either overweight or obese. The available scientific data show that workplaces are good for developing food-intake interventions for a healthier life. The present study aims to report the design, protocol and methodology for the evaluation of the food intake and physical activity patterns of the Uruguayan State Electrical Company (UTE) workers, as it is distributed across the whole country, and has established associations with overweight and obesity in order to establish institutional strategies to improve the situation. This study uses a population and a cross-sectional, randomized, representative sample of UTE workers with a precision of 3% and a confidence level of 95%. The considered anthropometric variables are weight, height, waist circumference, percentage of fat mass and percentage of visceral fat. A questionnaire on frequency of consumption of different foods and two 24-h dietary recalls (24-h DR) will be performed to evaluate the food intake. Accelerometry will be used to evaluate physical activity, and the International physical activity questionnaire (IPAQ) will be applied. Clinical data will be obtained from the UTE clinical charts. This is the first study of its kind that will be undertaken in Uruguay. It is registered under ClinicalTrials.gov Identifier nº NCT04509908.

## 1. Introduction

Noncommunicable diseases (NCDs) represent one of the major challenges of the 21st century for health and development. They represent the main cause of death globally. Forty-one million annual deaths are due to NCDs, 71% of deaths in the world. Of these, 15 million people die prematurely before 70 years of age. Most of these deaths are potentially preventable; the diseases last a long time and generally evolve slowly. The four main kinds of NCD are cardiovascular diseases (CVD) (heart attacks and strokes), cancer, chronic respiratory diseases (chronic obstructive pulmonary disease) and diabetes [1].

NCDs affect all ages and all regions and countries. These illnesses are usually associated with the older age groups; 36.6% of all deaths attributed to NCDs occur in people between 30 and 69 years of age. Children, adults, and the elderly are all vulnerable to the risk factors that produce NCDs, such as overweight and obesity due mainly to bad diets and physical inactivity, exposure to tobacco smoke, or inappropriate use of alcohol [1]. 

Obesity has reached epidemic proportions globally, and each year there are at least 2.8 million deaths due to obesity or overweight. In 2016, 39% of adults 18 years old or over were overweight (1900 million) and 13% were obese (650 million) [2]. Most of the world’s population lives in countries where overweight and obesity are already responsible for more deaths and more morbidity than underweight [3]. 

In the Second International Conference on Nutrition in 2014, the WHO recognized that some socioeconomic and environmental changes could affect changes in food intake and physical activity habits, leading to higher susceptibility to obesity and NCDs due to lifestyles that are increasingly sedentary, together with an increase in the consumption of food with high fat content, especially saturated and trans fats, sugars, and salt or sodium [4,5]. 

In recent decades, there have been important changes in Uruguay from the epidemiological, demographical and nutritional points of view that have had important consequences on the health of the population. NCDs are the main cause of morbimortality in Uruguay and the origin of most of the disabilities that people experience [6,7].

In September 2005, the ILO (International Labor Organization) published the report “Poor workplace nutrition hits workers’ health and productivity” [7]. This report refers to a study requested by the organization, “Food at work, workplace solutions for malnutrition, obesity and chronic diseases“, by Wanjek [8]. In this piece of work, it is affirmed that the workplace is an opportunity to develop nutritional interventions to improve health. It backs up this statement with the fact that workers spend half or more of their time at work, to which we can add long trips to and from their homes in many cases. 

According to Wanjek, unbalanced feeding during work can produce up to a 20% loss in productivity and important health problems. For the general community, the author proposes that businesses, through the promotion of healthy eating for their workers, have the opportunity to contribute to ameliorating the worst health problem most countries in the world face, i.e., NCDs (cardiovascular and tumors). 

Uruguay is in an advanced phase of nutrition transition where malnutrition problems due to excess prevail at all ages. In Uruguay, NCDs account for 60% of all health care costs and cause 70% of all mortality. Half of cardiovascular deaths occur in working-age people, with important consequences for family, society and organizations [6,7]. 

The hypothesis of our study, called IN-UTE, (Nutritional research at the National Administration of Power Plants and Electric Transmission UTE, Uruguay) is that the lifestyles and physical activity of the workers of UTE, as the company operates across the whole country, have an impact on the prevalence of overweight and obesity and other associated comorbidity factors. Indeed, the present study aims to report the design, protocol and methodology for the evaluation of the food intake and physical activity patterns of UTE workers and their associations with overweight and obesity in order to establish institutional strategies to improve the situation.

## 2. Methodology

### 2.1. Study Design and Sampe

This is a transversal observational study addressed to workers who started working between 1 January 2010 and 31 December 2017 at the UTE. The study variables are sociodemographic data, anthropometric parameters, intake of food and physical activity. This is complemented with study results of tests for risk factors for NCD.

#### 2.1.1. Inclusion and Exclusion Criteria 

Inclusion criteria. All the active UTE workers started between 1 January 2010 and 31 December 2017.

Exclusion criteria. Those who did not have a medical check-up on starting work and at least one follow-up between the first and second year of starting at the Company.

Workers who did not have at least two medical follow-ups 

Workers who did not have their weight and height registered on their charts in both follow-ups. 

#### 2.1.2. Sample 

The total population of the UTE is 6615 workers. The study population is 1964 (383 women and 1581 men) workers who started work in 2010, from when regular medical check-ups were performed in the Company. We later calculated a representative sample of this population with a confidence level of 95%, a statistical error (alfa) of 5%, and a precision of 3% to be subjected to accelerometry, 24 h dietary recall (24 h DR) and anthropometric measurement, following this equation: $$n\;=\frac{N*Z_{a}^{2}\;p*q}{d^{2}*\left(N-1\right)+Z_{a}^{2}*p*q}$$
*N* = population total*Z_a_*^2^ = 1.962 (if the confidence level is 95%)*p* = expected proportion (in this case 5% = 0.05)*q* = 1 − *p* (in this case 1 − 0.05 = 0.95)*d* = precision (in this case we want 3%)

The resulting sample size was 183 cases. In order to take into account possible losses, it was decided to increase this to 200 participants. The sample was randomly obtained, taking into account sex: 161 men (80.5%) and 39 women (19.5%). Each person was assigned a random number and a program of random number selection was used (http://www.generarnumerosaleatorios.com/, accessed on 1 July 2021) to select the subjects to be included in the sample. 

### 2.2. Ethical Aspects

The Research Ethics Committee of the University of Granada (Spain) reviewed and accepted the final protocol according to the ethical standards of the Declaration of Helsinki of 1964. It is coded as 1088/CEIH/2020 dated 3 March 2020. The study was subsequently registered at ClinicalTrials.gov (Protocol Registration and Results System—PRS—Receipt Release Date: NCT04509908).

Before any information is requested, the study will be explained to the participants. The refusal of a participants to provide a signed consent form or to complete any study phase will result in their exclusion. Participants of the study can stop the interviews at any moment, and their previously collected partial information would be excluded from any analysis in these cases.

Once the data from the medical histories and the surveys are obtained, they are anonymized for processing.

### 2.3. Survey Instruments

#### 2.3.1. Food Frequency Questionnaire (CFCA)

The CFCA is a semiquantitative form with 159 items, which is sent via web to the 1964 employees that comply with the criteria. This form takes about 30 min to complete. 

The STEPS form was used as a reference (progressive method of PAHO for the surveillance of the risk factors for chronic diseases (PAHO—STEPS) [9]. Validation of the adaptation was performed for the study population (Table 1).

#### 2.3.2. Twenty-Four Hour Dietary Recall (24-h DR)

The most commonly used method for quantitatively estimating food consumption and assessing the energy and nutrient intake is the 24-h DR [11,12]. In this method, participants recall everything they have eaten and drunk while at home or away from it, in the last 24 h (one day´s intake). In IN-UTE, we consider two non-consecutive recalls, the first during a face-to-face interview and the second during a ZOOM interview. 

The participants are not advised before the recall, nor when it will be carried out. A special questionnaire has been designed following EFSA (European Food Safety Authority) criteria [13]. This has been reviewed and modified in a pilot study (see below).

The participants are asked to recall what they have eaten in the last 24 h with domestic measurements using the Photographic Guide of Food Eaten in Spain [14]. 

The energy and nutrient intake will be estimated using the EvalFINUT software [15], As a food database, this uses the National Nutrient Database for Standard Reference (USDA) [16], with the addition of foods and preparations that are commonly used in Uruguay.

#### 2.3.3. Physical Activity Questionnaire

The questionnaire on physical activity is the IPAQ (International Physical Activity Questionnaires) validated for this population. It is administered together with the CFCA via the web to the 1964 employees. This section has approximately nine items and takes approximately 5 min to complete [17].

#### 2.3.4. Physical Activity

An Actigraph accelerometer (model GT3X) is placed on the non-dominant wrist for three consecutive days in those participants who have been selected in the population sample.

When the accelerometer is placed on the employee, the interviewer offers instructions for its use and places the gadget on the employee. These instructions are that the accelerometer should be placed on the non-dominant hand, all day and all night. It should not be removed at any time (except for contact with water).

After this period the employee goes to the workplace to return the accelerometer and the interviewer downloads the participant’s data.

The information obtained through this gadget is complemented with the subject’s sex, date of birth, height and weight.

The objective measurements of physical activity are carried out with the ActiGraph accelerometer, which registers the vertical movement, where the number of movements increases with the intensity of the activity. The accelerometer registers different periods of the day in which different levels of activity occur, i.e., different levels of “counts per minute” (cpm). This provides a measure of the frequency, intensity and duration of the physical activity and allows the activity to be classified as sedentary, light, moderate or vigorous [18]. 

For this study, it was decided that the minimum period to be included in the analysis is three days. The daily mean cpm for each participant is calculated as the weighted means based on the probability of use/non-use (for a minimum period of 8 h per day). 

The data obtained from the accelerometer is used to validate the physical activity questionnaire applied to the population sample and for the construction of a mathematical model to quantify the energy expenditure with different formulas.

The Energy Expenditure (EE) in this study is estimated using complementary measurements: objective method (accelerometer) and the self-questionnaire IPAQ. In the sample with accelerometer measurements, the EE is calculated as also the sum of static metabolic rate, RMR (Harris and Benedict’s formula) [19], and Freedson´s formula for physical activity for adults [20]. 

#### 2.3.5. Anthropometry

Anthropometric parameters are taken during the personal interview with the protocols of the International Society for Advancement of Kineanthropometry (ISAK) and quality procedures 

Height (cm): measured three times using a stadiometer (SECA 206, measuring range: 0 a 220 cm, graduations: 1mm/1/8 in).

Weight (kg): measured once (OMRON scales bf511).

Waist circumference (cm): measured three times with a measuring tape (SECA 201, measuring range: 0 a 220 cm, graduations: 1 mm).

The measurements take a total time of approximately 5 min. 

Assessment of body composition: body composition is determined as percentage of total body fat, lean mass, and visceral fat by prediction of bioelectric impedance analysis, only one measurement taken (OMROM bf 511 scales), total time 5 min. 

#### 2.3.6. Clinical Data

The data from clinical history are taken from the initial contact with the company and the one-year-later follow-up. 

These records are mainly on paper, so the necessary information is digitalized. The records are written by medical doctors specialized in occupational medicine. The data include: sociodemographic: age, sex, marital status, occupation, place of residence, educational level.

Personal history includes the diagnosis of any of the following diseases: diabetes, dyslipidemia, heart disease, arterial hypertension, oncological diseases, thyroid illnesses, sciatica, hyperuricemia, osteoporosis, bone fractures, chronic obstructive pulmonary disease, ulcers/gastritis, gallstones, diverticular disease, hepatitis and nephropathy. 

Family history includes hypertension, cardiovascular disease, diabetes, asthma, allergies, psychiatric history, addictions, neurological diseases, oncological diseases, obesity, autoimmune diseases, renal and thyroid diseases.

Job characteristics and risks: organization and division of tasks (rotating rests, rotating shifts, night shift, on call, customer service), work accidents, non-work accidents, prolonged absenteeism and its reasons.

### 2.4. Quality Controls

#### Pilot Study

In order to guarantee the best design and adaptation of the CFCA, the IPAC questionnaire and the 24-h DR to be applied, a pilot study was undertaken for these tools before the final validation.

For the pilot phase, 20 people of the same age group outside the company and 20 inside the company but not part of the study were interviewed and CFCA and IPAC applied.

The 24-h DR was required of five people outside the Company from the same age groups.

The objective and usefulness of this pilot study were to enhance the quality of the information through different procedures in the study and to detect limitations:✓ An adequate understanding of the different surveys is required to obtain trustworthy information.✓ The validation of the process of acquiring the data for each food’s intake, coding and treatment.✓ Usefulness and validity of the questionnaires in reaching goals.

Each stage of the study has strict quality controls.

Data obtained from the clinical charts. The operative coordinator of fieldwork supervises the quality and accuracy of the data weekly choosing a 10% random sample of the daily medical charts and checking that the data were accurate. At the end of the data collection, the main investigator chooses 20 random medical charts and verifies the information obtained.CFCA and IPAC. The nutritionist in charge of coordinating the fieldwork repeats the CFCA and IPAC in 10 employees, randomly chosen to verify that the information obtained is correct.24-h DR. This is applied by a professional nutritionist specialized in this methodology. Special diets are taken into consideration.Under-information and over-information of energy intake. Self-reporting of food consumption generally underestimates energy and nutrient intake. The assessment of errors in food consumption data base on under- or over-reporting of food intake is an important issue to be considered.Energy intake will be assessed using the cut-off points and the method proposed by Goldberg [21] and updated by Black [22]. BMR will be estimated through the Schofield equations [23], considering height, weight, age and sex. The specific cut-off points for the age and sex groups will be calculated considering the specific reference values and the coefficient of intra-individual variation for energy intake (EI), BMR, and physical activity, according to the method described by Nelson et al. [24] and Black [25].Accelerometry. The professional in charge of fieldwork has been specially trained in fitting the accelerometers at the University of Granada, Spain. The downloading of the information and its analysis is done by personnel from the Institute of Physical Education of the University of the Republic, who have training in this kind of analysis.

### 2.5. Statistical Analysis

All the above mentioned data will be entered into a database. The statistical treatment of the data will be carried out using the software STATA Statistics for Windows, Version 15.1.

A descriptive analysis of each variable will be performed. According to the distribution of each variable, the summary statistics to be used are median, ranges, means ± standard deviation (SD), percentiles, percentages for numerical variables, and absolute and relative frequencies (n and %) for categorical variables. This analysis will be stratified by sex and age.

To test differences between groups and within groups, the following tests may be used: Student’s *t*-test, chi-square test, Wilcoxon test, Mann–Whitney U-test, one-way ANOVA test and/or Kruskal–Wallis test, according to the distribution of the samples and the test statistics. The significance level will be set at *p* < 0.05, with a 95% confidence level. 

## 3. Outcome Measures

Dietary habits are estimated using the quantitative CFCA, and intake frequency values are converted into portions and grams of daily and weekly consumption for all food items. Scores are applied to indicate the diversity [26,27] and variety of food consumption in our study population, as we can see in EsNuPy [28].

Dietary patterns will be described using different methods, such as principal component analyses (PCA) and cluster analysis, among others [11,29]. To assess the overall dietary habits of the participants, the consumption frequency of all food items will be classified as meeting or not meeting the criteria of the dietary guidelines for the Uruguayan population [30] and the nutrition goals for the Uruguayan population.

### 3.1. Energy, and Macro- and Micronutrient Composition of the Diet

The 24-h DR is the technology used to transform food consumption into energy intake, water, macronutrient intake (carbohydrates, proteins, fats, fiber and micronutrient intake (vitamins and minerals)).

EVALFINUT is the software used to calculate mean daily intakes of energy and nutrients based on the food composition tables [31]. Data obtained will be evaluated using the dietary reference values of the Food Guide of the Uruguayan population, along with the Nutritional Objectives of the Uruguayan population [32].

### 3.2. Diet Quality

To assess overall diet/dietary pattern weighted by the different components of a healthy diet, “Diet/dietary quality indexes” are tools used to evaluate the level of adherence to a specified pattern or a set of recommendations in populations. For the IN-UTE study, the CFCA and the 24-h DR will be used to calculate different diet quality indexes, e.g., Update of the Healthy Eating Index [33].

### 3.3. Active and Sedentary Behavior Habits

All the activities picked up by the accelerometers are converted to energy expenditures using the metabolic (METy) according to the Adult Compendium of Physical Activities [34].

The calculations of energy cost are based on the METy value from the Youth Compendium, a measured or computed basal metabolic rate (BMR), and the duration of each specific activity: 

Energy cost (kcal) = METy × BMR (kcal/min) × duration (min), using the specific Schofield equation, the BMR is predicted for each adult [35].

The physical activity is measured based on the intensity of the activities and classified according to the latest recommendations of WHO (World Health Organization) guidelines as sedentary (≤1.5 METs), light-intensity activities (1.5–4 METs), moderate-intensity activities (4–7 METs), and vigorous-intensity activities (>7 METs) [32].

Participants daily physical activity will also be evaluated using the WHO recommendations on physical activity, sedentary behavior, and sleep for adults between 18 to 64 years old [36].

The energy balance will be calculated by comparing the energy intake obtained in the 24-h DR compared with the energy expenditure calculated from the basal metabolic rate (BMR), according to Schofield equations, plus the energy derived from physical activity obtained through the accelerometry.

### 3.4. Body Mass Index

In the IN-UTE study, weight and height are taken from the UTE medical history of the population; in the sample of 200 participants anthropometry of weight, height, waist circumference and estimation of %fat mass, %lean mass and %visceral fat are performed as discussed in the section on instruments. Evaluation of the indicator BMI are performed at an individual level, and subjects are classified according to the WHO classification [37] and waist circumference by sex [38].

### 3.5. Sociodemographic Characteristis

Factors that influence food choice are many, among them diet quality, budget, resources, household structure, and the availability of different foods (at a group level). At an individual level also, it is important to take into account taste preferences, nutritional knowledge, food attitudes and identity, health motivations, and habitual behavior [39].

The education is classified according to the International Standards Classification of Education [40]. 

### 3.6. Work Characteristics

The work characteristics of the participants are taken into account. We consider whether the work is active or inactive based on the tasks performed daily.

Work rotating schedules, 24-h shifts or night shifts are also taken into account given the importance in nutritional status and chronic diseases [36,41,42,43].

## 4. Discussion

The main objective of the study is to provide information to the company in order to be able to design useful and appropriate health interventions.

For the first time in Uruguay, these instruments will be used to measure physical activity and dietary intake for workers, having already been used in several studies [44,45,46,47].

Due to the COVID pandemic some adaptations are expected but are considered in the study protocol. 

## 5. Potential Intervention That Can Be Implemented with the Data from the Study

The results of the study can provide a baseline in order to evaluate interventions of the following kind:

1. Outpatient clinics for a healthier life with personalized nutritional and physical activity advice and follow-up in the company.

2. Establishment of a company policy on health promotion, especially with respect to nutrition and physical activity. This would include web-based promotion, using the company web to distribute useful material permanently.

## 6. Conclusions

The strong innovative points of the design, protocol and methodology used in this study are:-It is the first dietary survey in a public Company of this size and that specifically takes into account the paradigm of “energetic balance” at a population level.-It involves a population of 1964 employees between 18 and 55 years of age.-It is the first survey carried out on the same subjects who have a medical history on entering the company, a follow-up one year after starting work, and a further follow up which, will provide the opportunity to collect information on diet, physical activity, anthropometry and body composition.-It uses a web-based questionnaire on frequency of consumption which will be applied for the first time to employees of the company.-It includes a precise quantification of the physical activity of the employees of the state electrical company (combined use of self-reported questionnaires and objective accelerometers) to avoid the key problem of under-reporting and to assess factors such as type, duration, quantity and intensity, which are rarely reported in nutritional population surveys.

The main difficulties that we will encounter are the worldwide pandemic and the fact that the country is going through a critical period, with restrictions on mobility, work at home, etc. 

Another difficulty is that the subjects are employees of the company and therefore, the data on alcohol and drug consumption may be biased. 

This investigation will study overweight and obesity in-depth and their associated factors in a working population. This study will be the basis for the design of recommendations and intervention strategies for the employees of the UTE company. 

## Figures and Tables

**Table 1 nutrients-13-03545-t001:** Main foods by group and subgroup, and their reference servings used in CFCA in the IN-UTE study.

FOOD GROUPS	
Dairy products: Milk whole or fat free (200 mL) Vegetable drinks (200 mL) Yogurt whole or fat free (125 g) Fresh cheese (15–30 g) and cured cheese (15 g) Sugared dairy dessert (125 g) and ice cream (120 g)	Eggs, meat, and fish: Eggs (64 g) Chicken, lean meat, medium-fat meat, fatty pork, lamb meat (120 g) Cooked ham, sausages, other processed meats (15 g) Fish: white fish (55 g), bluefish (55 g), fresh seafood (90–100 g), and canned fish (25 g)
Oils and fats: Olive oil (10 g) Other vegetable oils (10 g) Margarine and butter (12 g)	Bakery products: Cookies and cakes made with butter or oil (25–50 g) Homemade bakery products (50 g) Sugary cocoa (5 g), nougat (20 g), and shortbread cookies (45 g)
Vegetables and potatoes: All kind of vegetables (150 g) Potatoes and sweet potatoes (100 g)	Miscellaneous: Industrial pre-cooked products (25–50 g) Homemade pre-cooked products (25–50 g) Sauces (8 g) Condiments (0.25 g) Sugar, honey, and marmalade (10 g) Low- or no-calorie sweetener (0.05 g) Snacks (fried snacks, 50 g) Sweets, chocolates (10 g)
Fruits and nuts: All kinds of fresh fruits (100–200 g) Olives (20 g) Fruits with juice and fruits with syrup (100 g) Dried fruits (30 g)	Beverages: Sugar-sweetened carbonated beverages or low-calorie carbonated beverages (200 mL) Natural juices and commercial juices (200 mL) Mixtures of fruit juice and milk, sugar-sweetened or low-calorie sweetened (200 mL)
Legumes, cereals, and pasta: Legumes (40 g) Bread (20–25 g) Breakfast cereals with or without sugar (30 g) Rice, cornmeal, quinoa and pasta (60 g) Stuffed pasta (100 g)	Alcoholic beverages: Beer (200 mL) Wine (150 mL) White drinks (50 mL)

Table adapted from the EsNuPi study based on the adult population and eating habits of the Uruguayan population [10].

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
