# Peer review of "Study of Food Intake and Physical Activity Patterns in the Working Population of the Uruguayan State Electrical Company (UTE): Design, Protocol and Methodology"

_nutrients, 2021, doi:10.3390/nu13103545_

Round 1
Reviewer 1 Report
Medina-Vadora and colleagues want to evaluate the food intake and physical activity patterns of the Uruguayan State Electrical Company (UTE) workers, which is distributed in the whole country and the associations with overweight and obesity to establish institutational strategies to improve the situation.
Why the study is called IN-UTE? What IN stands for?
In the abstract the study is called "a cross-sectional, randomized, representative ..." What is randomized in this study? There is no information in the methods part about the randomization! Please describe in detail and explain.
It is not clear whether the study is already being carried out or is still being planned - different times are used over and over again.
Introduction
Citation #1 and #2 are the same
In general, it has to be mentioned that the introduction was largely taken over from a homepage and largely consists of these citations and not primary scientific literature sources is cited.
L85, what ILO stands for? Please explain.
Sample calculation:
In the total population of the UTE there are 75% men - in the resulting sample size the study team will randomly recruite 80,5%, please explain why?
How will you select the 200 cases from the total population?
L176: Registration Number is not correct
Citation #19 is missed, please cite a reproducable source
The present manuscript is confusing and unstructured. The only figure is not explanatory but raises even more questions - including typing errors. There is no structured, clear structure, perhaps supplemented with some explanatory graphics. The discussion part is very long - far too long for a study design. I would like to mention that If the results should be integrated into the work immediately, the design is weak.
Author Response
Please read attached file.
Regards.
Msc. Mercedes Medina Vadora

Reviewer 2 Report
This research article aims to examine the association between food intake and physical activity pattern in working population in the Uruguayan state electrical company.
The objective of conducting this research is persuasive but too simple. Several suggestions should be addressed.
-Figure 1 needs to be more specifically presented
-Food Frequency questionnaire needs to be provided with a new Table, which will help readers for understanding the FFQ form consisting of food items and eating frequency.
-As far as I understand, this study has one objective but why this study divided into primary and secondary outcome measures in section 3. The aim of study needs to be addressed in a specific way.
-Physical activity questionnaire should be provided for readers’ understanding.
Author Response
Please see the attachment.
Regards.
Msc. Mercedes Medina Vadora

Round 2
Reviewer 1 Report
Many thanks for the detailed answers to the queries and the revision of the manuscript.
Reviewer 2 Report
Authors addressed commnets in an appropriate way.